# Progressive and Coordinated Mobilization of the Skeletal Muscle Niche throughout Tissue Repair Revealed by Single-Cell Proteomic Analysis

**DOI:** 10.3390/cells10040744

**Published:** 2021-03-28

**Authors:** Matthew Borok, Nathalie Didier, Francesca Gattazzo, Teoman Ozturk, Aurelien Corneau, Helene Rouard, Frederic Relaix

**Affiliations:** 1Université Paris Est Creteil, INSERM, IMRB, F-94010 Creteil, France; matthew_borok@inserm.fr (M.B.); nathalie.didier@inserm.fr (N.D.); francesca.gattazzo@inserm.fr (F.G.); teoman.ozturk@inserm.fr (T.O.); helene.rouard@inserm.fr (H.R.); 2EFS, IMRB, F-94010 Creteil, France; 3Sorbonne Université, UMS030 LUMIC, Plateforme CyPS, Hôpital Pitié-Salpêtrière, F-75013 Paris, France; aurelien.corneau@inserm.fr; 4ENVA, IMRB, F-94700 Maisons-Alfort, France; 5AP-HP, Hôpitaux Universitaires Henri Mondor & Centre de Référence des Maladies Neuromusculaires GNMH, F-94010 Creteil, France

**Keywords:** skeletal muscle, muscle stem cells, satellite cells, regeneration, muscle niche, CyTOF

## Abstract

*Background***:** Skeletal muscle is one of the only mammalian tissues capable of rapid and efficient regeneration after trauma or in pathological conditions. Skeletal muscle regeneration is driven by the muscle satellite cells, the stem cell population in interaction with their niche. Upon injury, muscle fibers undergo necrosis and muscle stem cells activate, proliferate and fuse to form new myofibers. In addition to myogenic cell populations, interaction with other cell types such as inflammatory cells, mesenchymal (fibroadipogenic progenitors—FAPs, pericytes) and vascular (endothelial) lineages are important for efficient muscle repair. While the role of the distinct populations involved in skeletal muscle regeneration is well characterized, the quantitative changes in the muscle stem cell and niche during the regeneration process remain poorly characterized. *Methods***:** We have used mass cytometry to follow the main muscle cell types (muscle stem cells, vascular, mesenchymal and immune cell lineages) during early activation and over the course of muscle regeneration at D0, D2, D5 and D7 compared with uninjured muscles. *Results***:** Early activation induces a number of rapid changes in the proteome of multiple cell types. Following the induction of damage, we observe a drastic loss of myogenic, vascular and mesenchymal cell lineages while immune cells invade the damaged tissue to clear debris and promote muscle repair. Immune cells constitute up to 80% of the mononuclear cells 5 days post-injury. We show that muscle stem cells are quickly activated in order to form new myofibers and reconstitute the quiescent muscle stem cell pool. In addition, our study provides a quantitative analysis of the various myogenic populations during muscle repair. *Conclusions*: We have developed a mass cytometry panel to investigate the dynamic nature of muscle regeneration at a single-cell level. Using our panel, we have identified early changes in the proteome of stressed satellite and niche cells. We have also quantified changes in the major cell types of skeletal muscle during regeneration and analyzed myogenic transcription factor expression in satellite cells throughout this process. Our results highlight the progressive dynamic shifts in cell populations and the distinct states of muscle stem cells adopted during skeletal muscle regeneration. Our findings give a deeper understanding of the cellular and molecular aspects of muscle regeneration.

## 1. Background

Skeletal muscle is an essential tissue for locomotion, body heat and metabolism. It is also one of the most regenerative tissues found in mammals, able to regenerate in response to a variety of traumas and pathological conditions. Following injury by injection of a myotoxin, such as the snake venom cardiotoxin or chemical reagents such as Barium Chloride (BaCl2), the mouse tibialis anterior (TA) muscle can fully regenerate in one month and is capable of doing this repetitively [1,2]. This regenerative capacity is largely dependent upon the satellite cell, the postnatal skeletal muscle stem cell. In adults, satellite cells are quiescent and located under the basal lamina in close contact with myofibers. They express a set of specific markers such as the transcription factor PAX7 or the adhesion proteins ITGA7 or M-CADHERIN [3,4]. Upon injury, the muscle satellite cells activate, proliferate and differentiate to produce new myofibers [5]. A subpopulation of the activated satellite cells will self-renew to replenish the stem cell pool for future needs.

While it is challenging to follow the individual differentiation paths of satellite cells in vivo [6], ex vivo culture of single myofibers provides a model where activated satellite cells first upregulate the transcription factor MYOD. PAX7+MYOD+ cells can either downregulate MYOD to return to a quiescent state or engage in the differentiation program by downregulating PAX7 and begin expression of the transcription factor Myogenin (MYOG) [5]. Differentiated MYOG+ cells are then able to fuse to existing myofibers or seed the formation of new myofibers.

In vivo, satellite cells and myofibers are surrounded by a number of other cell types. In the past decade, increasing attention has been given to the support cells of the skeletal muscle niche: endothelial cells, fibroadipogenic progenitors (FAPs), pericytes and immune cells, all of which interact with satellite cells and impact their regenerative potential [7,8,9,10,11]. During regeneration, these populations go through rapid changes, with inflammatory macrophages recruited immediately to clear dying fibers. These macrophages differentiate into anti-inflammatory macrophages within several days and contribute to the formation of new myofibers [11]. A large proportion of endothelial cells die in the early stages of regeneration, while the FAPs undergo an expansion two days following injury [2,12].

With the advent of new techniques like cytometry by time-of-flight (CyTOF) and single cell RNA-seq, we are learning that even more cell types exist in the skeletal muscle niche [13,14,15,16,17,18]. However, the preparation of skeletal muscle for these analyses necessitates a lengthy digest of the tissue, which is now known to have transcriptional consequences on satellite cells, leading to an early activation state [19,20]. Only two hours of digestion was sufficient to induce thousands of significant transcriptional changes in satellite cells, compared to paraformaldehyde-fixed quiescent satellite cells [19,21].

Here, we designed a CyTOF panel to follow the different skeletal muscle cell populations in early activation phase and different stages of regeneration. Our analysis showed rapid changes in the proteome of early activated endothelial, FAP and satellite cells. The changes in cell populations we observed during regeneration are in good agreement with similar studies using scRNA-seq and CyTOF [17,18]. However, our inclusion of a myogenic lineage tracer, and antibodies for myogenic transcription factors provide a more nuanced picture of satellite cell differentiation in vivo.

## 2. Methods

### 2.1. Mouse Strains

The *Pax7nGFP* line has been described previously [22]. Animals were handled according to national and European community guidelines, and protocols were approved by the ethics committee at the French Ministry (Project No: 13696).

### 2.2. Induction of Muscle Damage

Mice were anesthetized by intraperitoneal injection of ketamine-xylazine solution. Tibialis anterior (TA) muscles of anesthetized mice were injected with 50 μL of 0.6% BaCl_2_.

### 2.3. Collection of Muscle Cells and CyTOF Antibody Staining

Preparation of native (T0) and digested (T3) muscle was performed as previously described [19]. For D0 (uninjured), D2, D5 and D7 samples, the same protocol was used with the following modifications: 5 mL of dissociation media (Collagenase A and Dispase) was used for each sample and TAs were incubated at 37 °C shaking for 30 min, before a quick spin and filtration of the liquid portion through a 40 micron filter. Next, 500 µL of FBS was added to the filtrate to block enzyme activity. Then, 5 additional mL of dissociation media was added to the pellet and samples were returned to the shaking 37 °C incubator for 30–90 min. The number of TAs used per sample was 4 at 2DPI, 6 at 5DPI, and 12 at 7DPI. For D0, 20 mL of dissociation media was used to digest 24–34 TAs per sample.

Following filtration, cells were stained using the Transcription Factor Phospho (TFP) Buffer Set (BD Biosciences, San Jose, CA, USA). Briefly, cells were washed with the TFP Wash Buffer, then incubated with Cell-ID Cisplatin (Fluidigm, San Francisco, CA, USA) for 5 min. Cells were washed with TFP Wash Buffer and then incubated with cell surface antibodies for 45 min. Cells were then fixed and permeabilized, before addition of intracellular antibodies for an additional 45 min. Cells were washed once more with TFP buffer, then incubated in Cell-ID Intercalator (Fluidigm) in 2% PFA in polypropylene tubes. Stained cells were frozen at −80 °C. The next day the cells were thawed and washed in Stain Buffer (BD Biosciences) and brought to the mass cytometer for analysis.

For T0 and T3 samples were frozen in 10% DMSO, 90% FBS at −80 °C. Staining was performed as described above after thawing.

### 2.4. Processing of CyTOF samples

A Helios mass cytometer was used as previously described to analyze stained samples [16].

### 2.5. CyTOF Data Analysis

CyTOF data were analyzed using Cytobank. Visualization of t-Distributed Stochastic Neighbor Embedding (viSNE) analysis was performed with 7000 iterations. Manual gating was used to quantify distinct cell populations.

### 2.6. Single TA Muscle Dissociation and Immunostaining for Flow Cytometry Analysis

Single TA muscles from adult mice were carefully dissected and finely minced in HBSS (ThermoFisher, Grand Island, NY, USA). Minced TA were dissociated in 3 mL of a solution containing Collagenase A (2 mg/mL, Roche, San Francisco, California, USA), Dispase II (3 mg/mL, Roche), DNase I (10 µg/mL, Roche), 0.4 mM CaCl_2_ (Sigma, St. Louis, MO, USA) and 5 mM MgCl_2_ (Sigma), for 30 min (2 DPI) to 40 min (5 and 7 DPI) at 37°C with agitation every 5 min. After, 20 min of digestion, 2 mL of the digestion solution was recovered, diluted in 10 mL of HBSS added with 0.2% Bovine Serum Albumin (HBSS/BSA) and kept on ice. Then, 2 mL of fresh digestion solution was added to the remaining digestion tube. At the end of the digestion, the totality of the mononucleated cells obtained was washed with HBSS/BSA and filtered through 40-µm nylon filters (Corning, Corning, NY, USA). Red blood cells were lysed in 1 mL of Red Blood Cell lysing buffer (BD Biosciences) for 10 min. Cells were washed with HBSS, transferred to flow cytometry collection tube and labeled with 1 mL of Fixable Viability Stain FVS780 (1:1000, BD Biosciences) in HBSS for 10 min. Cells were washed with HBSS, and then fixed and permeabilized in 500 µL of Fixation solution (Transcription buffer set, BD Biosciences) for 45 min at +4 °C, in accordance with manufacturer’s instructions. Fixed cells were washed with the Permeabilization/Wash buffer (Transcription buffer set, BD Pharmingen) and proceeded for nuclear immunostaining as previously reported [23]. The following mix of conjugated antibodies was used: mouse anti-PAX3/7-AF647 (B-5) (Santa Cruz, Dallas, TX, USA #sc-365843), mouse anti-MYOD-PE (BD Biosciences, #554130, custom made coupling by manufacturer), mouse anti-MYOG-AF488 (5FD) (Santa Cruz, #sc-52903), mouse anti-KI67-bV421 (BD Biosciences, #562899). Specificity of the staining was verified using isotypic controls for each antibody [23]. Cells were washed with the Permeabilization/Wash solution, re-suspended in HBSS and transferred in BD Trucount^TM^ Absolute Counting Tubes (BD Biosciences, #340334) before being analyzed using a BD FACS Canto^TM^ Flow Cytometer (BD Biosciences). Acquisitions were performed on 20,000 beads.

### 2.7. Statistical Analysis

Results are represented as mean ± sem. Statistical analysis were performed using GraphPad Prism software. Statistical significance was determined by One and Two-Way ANOVA tests followed by Tukey’s multiple comparisons test. *p* < 0.05 was considered significant (* *p* < 0.05, ** *p* < 0.01, *** *p* < 0.001, **** *p* < 0.0001).

## 3. Results

### 3.1. Establishing a Single Cell Proteomic Panel for CyTOF Analysis of Skeletal Muscle

We have previously shown that skeletal muscle tissue dissociation leads to early activation of muscle stem cells [19], but the impact of dissociation for other lineages has not been evaluated. In order to investigate the proteome modification of myogenic and niche cells during dissociation-induced early activation and regeneration, we designed a panel of 42 antibodies (Table 1), including 18 that recognize surface markers for the major cell types in the niche, including CD31 for endothelial cells, NG2 for pericytes, PDGFRa for FAPS, LY6G for granulocytes, F4/80 for macrophages, CD3 for T cells and M-CADHERIN for satellite cells. The panel also contained antibodies for different signaling pathways, including pSTAT3 and pS6 and markers of cell division, including KI67 and pHH3.

### 3.2. Single Cell Proteomic Analysis Reveals Rapid Modifications During Dissociation.

As a model of early activation of the skeletal muscle cells, we compared fixed (T0) and activated (T3) *Pax7-nGFP* mouse muscles by single cell mass cytometry as described previously [19]. The *Pax7-nGFP* line has integrated a BAC containing over 100 kb of the Pax7 locus with GFP knocked into the beginning of the *Pax7* gene [22]. Thus, GFP can serve as a marker of *Pax7* locus activity, and through perdurance of the protein, the myogenic lineage. Using visualization of t-Distributed Stochastic Neighbor Embedding (viSNE), we generated a representation of the various cell populations of the skeletal muscle niche in each condition (Figure 1A). In agreement with previous studies, we identified a muscle-specific distribution of different cell types, including satellite cells, endothelial cells, FAPs and immune cells [16,17]. We observed small changes in the proportion of endothelial cells (24% in T0 and 32% in T3), FAPs (1% in T0 and 2.5% in T3) and immune cells (29% in T0 and 20% in T3). This may reflect cellular loss or enrichments during dissociation, or that enzymatic dissociation affects the stability and exposure of specific surface antigens.

Strikingly, our panel was able to detect protein changes in multiple cell types upon dissociation, demonstrating that early response to dissociation is not restricted to the muscle stem cells. STAT and mTOR signaling, marked by phospho-STAT3 and phospho-S6, respectively, were rapidly activated in endothelial (46% and 30%) and FAP cells (37% and 96%). Interestingly, STAT signaling was more present than mTOR signaling in the endothelial cells, while mTOR signaling was dominant in the FAPs (Figure 1B,C).

Activated satellite cells also had high levels of phospho-STAT3 and phospho-S6 (16% and 61%, respectively). STAT3 signaling has been previously implicated in response to stress in multiple cell types of the vasculature [24]. Furthermore pSTAT3 directly binds and promotes transcription of *c-Fos*, *Socs3*, *Jun*, and *c-Myc*, [25,26,27,28], all components of the “stress core” we recently showed is upregulated in many early activated cell types from distinct tissues [20]. Activation of mTORC1 signaling was previously shown to be an indicator of cellular stress and priming for activation in satellite cells and FAPs [29,30] (Figure 1D).

Furthermore, activation led to a large decrease in the GFP+PAX3/7+ population, consistent with the drop seen in *Pax7* mRNA, one of the most significantly affected genes in our previous RNA-seq analysis [19]. This most likely reflects loss of the PAX7 protein in the population, as the GFP+ITGA7+ population actually increases slightly.

### 3.3. Muscle Niche Lineages Show Drastic Changes During Skeletal Muscle Regeneration

In addition to dissociation-induced changes, we were interested in the changes in the muscle stem cell niche that occur during injury and regeneration. For this purpose, we injured the TA muscles of *Pax7nGFP* mice with a single injection of BaCl_2_, and performed CyTOF analysis 2, 5 and 7 days following the injury (Figure 2A). Due to the large number of TAs required for each sample and the potential loss of material with the fix protocol, we opted to collect these muscles with the traditional digestion protocol. We thus also collected uninjured TA muscles as controls for these CyTOF analyses. Two days following injury (D2), we observed that 60% of the cells in our preparations were immune cells, including granulocytes, pro-inflammatory M1 and anti-inflammatory M2 macrophages (Figure 2B,C). In the non-immune compartment, we saw a sharp decrease in endothelial cells, with smaller drops in FAPs, pericytes and myogenic cells (Figure 2D). Relative proportions of myogenic, FAP and pericyte cells all peaked 5 days after injury, coinciding with full phenotypic shift of macrophages from M1 to M2. At day 7, the proportion of immune cells begins to decline and is still largely made up of M2 macrophages. The relative proportion of endothelial cells is still increasing to reach uninjured levels, while the proportions of FAPs, pericytes and myogenic cells are still elevated relative to the uninjured condition. The changes in different cell populations were in good agreement with previous reports using histology, FACS, scRNA-seq and CyTOF [2,17,18].

### 3.4. Dynamic Modification of the Myogenic Population Proteome During Regeneration

To investigate the changing myogenic cell proteome, we performed viSNE analysis on GFP-gated cells, revealing a well-synchronized pattern of protein expression (Figure 3). The GFP+ cells of uninjured TA muscles uniformly express PAX3/7. While this is at odds with our earlier results on digested muscle, it is important to note that the viSNE plots indicate relative amounts of proteins across samples, so while PAX3/7 is reduced by the digest procedure, uninjured muscle still has high levels of the proteins relative to subsequent time points. Two days following injury the GFP+ cells express the cyclin-dependent kinase inhibitors P21 and P57, and MYOG, with almost undetectable levels of PAX3/7. Thus it seems that the majority of the quiescent satellite cell pool inhibits proliferation and undergoes terminal differentiation and fusion with existing muscle fibers at this stage-in good agreement with loss-of-function studies of Notch signaling in satellite cells [31,32]. We recently demonstrated that P57 is essential in satellite cells for muscle regeneration and P21 and P57 have essential, redundant functions in prenatal myogenesis [33,34]. By day five, two distinct groups of myogenic cells are evident. The first, major population is expressing relatively high levels of PAX3/7 and P57, while the second, minor population is expressing MYOG and P57, indicating this population is transiting to terminal differentiation. Based on KI67 staining, it appears that satellite cells begin proliferating between day two and day five following injury, with many cells still cycling at seven days.

In order to further investigate the satellite cell profile in single TAs, we performed a quantitative analysis of myogenic cells by flow cytometry using antibodies for PAX7, MYOD, MYOG and KI67 (Figure 4, Appendix A) as reported previously [23]. This analysis confirmed that the total number of PAX7+ cells/TA was drastically reduced two days following BaCl_2_ injury. We observed a decrease of 82% of the total PAX7+ cells compared to uninjured muscles (Figure 4A), supporting previous observations that the regeneration of the entire muscle can be achieved from a small number of satellite cells [35]. However, this may also be due to early activation, rather than loss of the satellite cells. At this early stage, the majority (86%) of the remaining PAX7+ cells were still in a quiescent state (Figure 4B,C). Afterwards, at D5 the number of PAX7+ cells significantly increased and 38% of the cells were cycling. At 7 days post-injury, the number of PAXX7+ cells/TA stabilized whereas the proportion of cycling PAX7+ cells was reduced compared to D5. While the total number of MYOD+ cells/TA did not vary significantly at the different time points, we observed a significant increase in the number of MYOG+ cells/TA at D2 compared to uninjured control (Figure 4D,E), in agreement with CYTOF analysis. The number of differentiating MYOG+ cells peaked at D5 and then strongly decreased by D7.

In addition, we assessed the repartition of the total myogenic cells (TM) at each time point, defined as the sum of the cells expressing at least one the marker (PAX7, or MYOD or MYOG) (Figure 4F, Appendix A). At D2, the majority of the myogenic cells were committed PAX7−MYOD+ cells or differentiating MYOG+ cells, whereas PAX7+ cells represented only 30% of the cells, suggesting that upon BaCl_2_ injury a subset of satellite cells rapidly progress in the myogenic differentiation program. At D5, consistent with the strong increase of PAX7+ cell number (Figure 4B), we observed a reduction in the proportion of MYOD+ cells associated with an increase in the proportion of PAX7+ cells. At D7, the proportion of differentiating MYOG+ cells decreased, and PAX7+ cells represented 87% of the total myogenic cells.

Overall, these analyses showed that the small pool of quiescent PAX7+ cells remaining after BaCl_2_ injury, actively amplified between D2 and D5. In the meantime, a subset of the cells committed to differentiation and then fused to repair or generate new myofibers while another subset self-renewed to replenish the pool of quiescent SCs. At D7, the amplification/differentiation phase of PAX7+ cells started to decline suggesting that at this stage myofiber repair process was almost completed.

## 4. Discussion

Our data highlight the sensitivity of the satellite cell niche proteome to standard dissociation protocols, and the exquisite fine-tuning of PAX7 levels during regeneration. Previous studies had shown that within the 2 h of the digestion protocol Pax7 transcripts are reduced by 10–20 fold [19]. This study shows that PAX3/7 protein is also quickly lost during digestion, suggesting a robust post-translational level of regulation. This may involve the ubiquitinase Nedd4, as it was previously shown to regulate PAX7 protein abundance [36]. Digestion also induced pSTAT3 signaling in endothelial cells and FAPs, which could have important implications for the interpretation of FAP heterogeneity [37]. Induction of these signaling pathways is also likely linked to a core set of genes upregulated in activated cells of many different tissues [20].

While complete muscle regeneration takes at least one month, our analysis reveals that many drastic changes in terms of cellular components of the niche occur in the first week following injury (Figure 5). In addition to this, we have found the large proportion of immune cells in injured muscles to be a useful indication of injury efficiency.

Our quantitative analysis of skeletal muscle satellite cells during regeneration shows that the PAX7+ cell population decreases by 82% two days following injury in our experimental conditions. This demonstrates that satellite cell loss and early activation significantly reduce the quiescent muscle stem cell pool, without impairing muscle regeneration. While previous work has suggested that 80% ablation of satellite cells significantly hinders regeneration [8,35,38,39], this may also be due to a negative impact of diphtheria toxin on the function of surviving satellite cells.

Finally, our study shows the utility of a CyTOF panel to track the major cell types of the niche and their behavior during regeneration. In addition, the FACS panel we describe is a rapid and relatively inexpensive way to assess the differentiation status of satellite cells at different stages of injury. Both techniques can be used in parallel in the future to assess regenerative phenotypes in different mutants with different models of injury.

## Figures and Tables

**Figure 1 cells-10-00744-f001:**
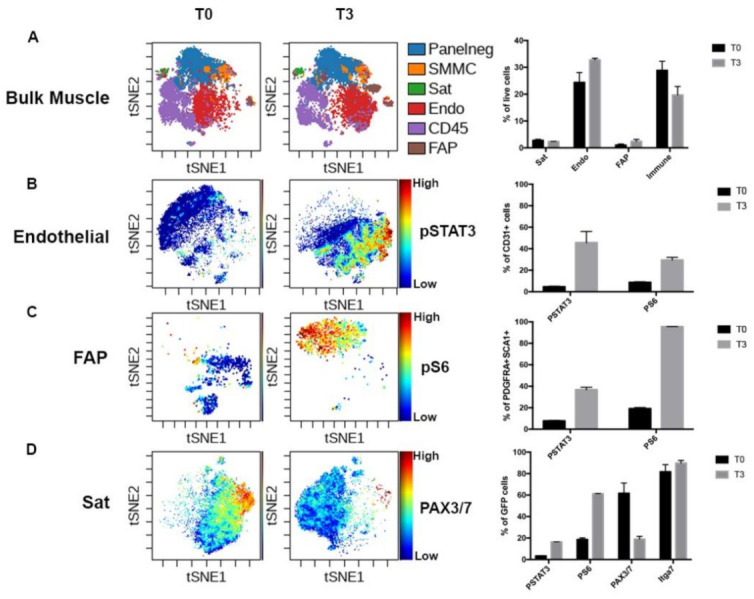
Early activation of skeletal muscle modifies the niche proteome. (**A**) Visualization of t-Distributed Stochastic Neighbor Embedding (viSNE) plot of single cell preparations of whole-body musculature with immediate fixation (T0) or two hours of digestion (T3). Manual gates were applied for the coloration of different populations. Ungated: cells not captured by the following manual gates, Panelneg: cells not captured by manual gating of standard skeletal muscle cell markers, likely containing glial and tenogenic cells (Giordani et al. 2019), SMMC: smooth muscle mesenchymal cells (ITGA7+ VCAM−) Sat: MuSC (Pax7− progenitors (PDGFRα+ SCA1+). Quantification by manual gating at right. *n* = 2 (**B**) viSNE analysis of endothelial cells with pSTAT3 levels indicated and quantification by manual gating at right. (**C**) viSNE analysis of FAPs with pS6 levels indicated and quantification by manual gating at right. (**D**) viSNE analysis of GFP+ cells with PAX3/7 levels indicated and quantification by manual gating at right. Low expression-blue, high expression-red.

**Figure 2 cells-10-00744-f002:**
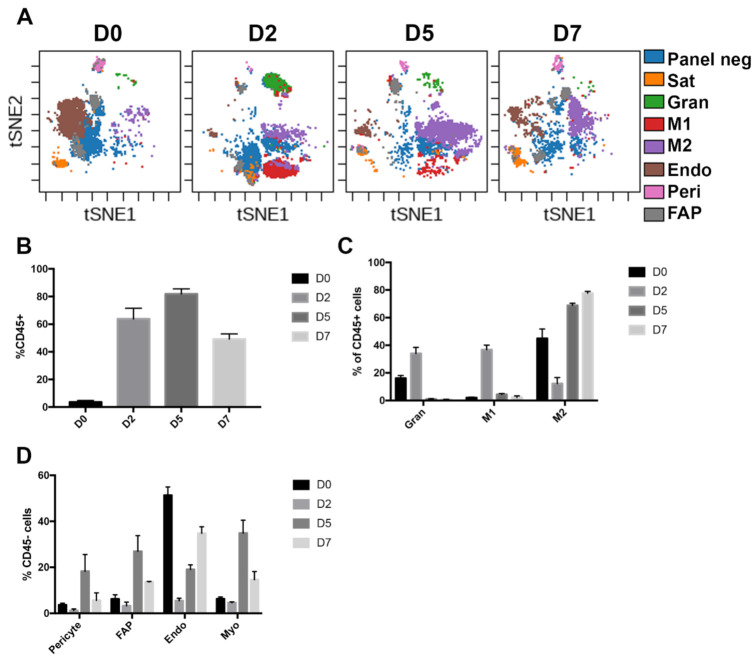
The cellular makeup of skeletal muscle is reshuffled following injury. (**A**) viSNE plot of regenerating TA muscle cell types before and following BaCl2 injury. Panelneg: cells not captured by manual gating of standard skeletal muscle cell markers, Myo: myogenic cells (Pax7-nGFP+), Endo: endothelial cells (CD31+), Peri: pericytes (PDGFRβ+ SCA1−), M1: type 1 macrophages (F4/80+ LY6C+), M2: type 2 macrophages (F4/80+ LY6C−), FAP: fibro-adipogenic progenitors (PDGFRα+ SCA1+), Gran: granulocytes (LY6G+). *n* = 3 for D0, D2, D5, *n* = 2 for D7. (**B**) Quantitation of immune cell proportion in resting and regenerating muscle. (**C**) Makeup of the immune compartment at different time points. (**D**) Makeup of the non-immune compartment at different time points.

**Figure 3 cells-10-00744-f003:**
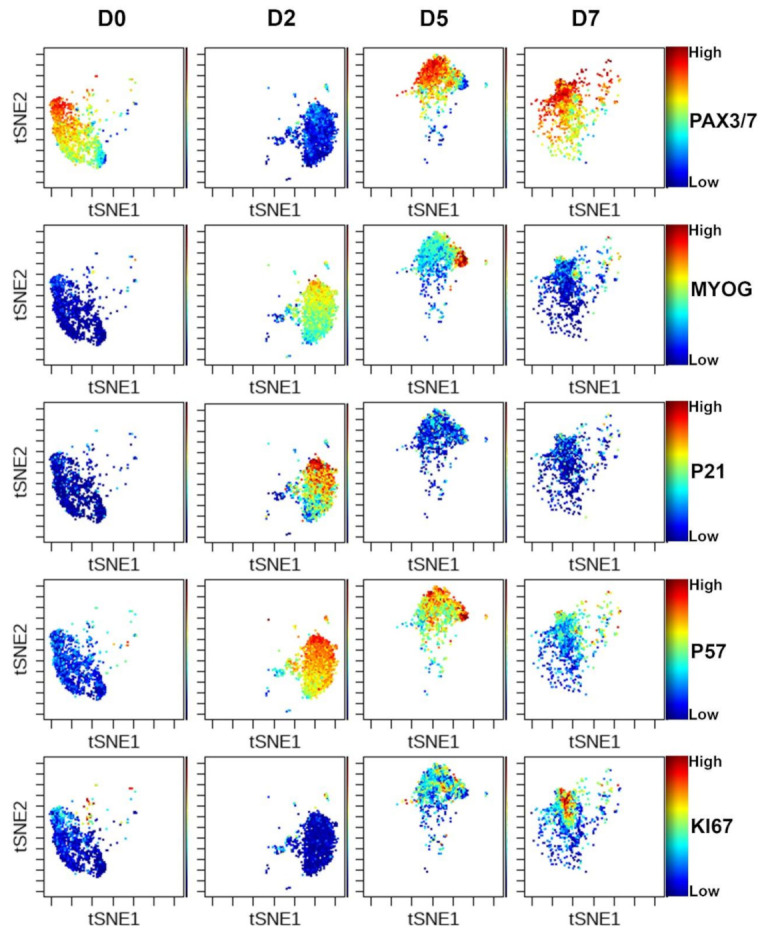
Cyclic activation of muscle stem cells during injury. viSNE analysis of GFP+ cells at each time point with several markers shown. Low expression-blue, high expression red.

**Figure 4 cells-10-00744-f004:**
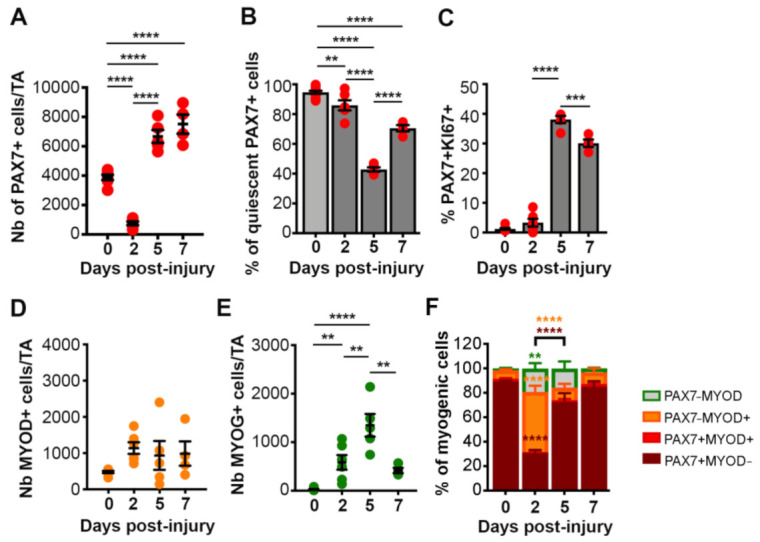
Quantitative FACS analysis of mononucleated cells from single TAs. Graphs showing the number of PAX7+ cells/TA (**A**), the % of quiescent PAX7+ cells (defined as MYOG−MYOD−Ki67−PAX7+ cells) (**B**), the % of KI67+PAX7+ cells (**C**), the number of MYOD+ cells/TA (**D**) and MYOG+ cells/TA (**E**), and the proportions of myogenic cells expressing PAX7 and MYOD (**F**). Data are represented as the mean ±sem of minimum 4 isolated TAs. One-way ANOVA or Two-way ANOVA (**F**) with ** *p* < 0.01, *** *p* < 0.001 and **** *p* < 0.0001.

**Figure 5 cells-10-00744-f005:**
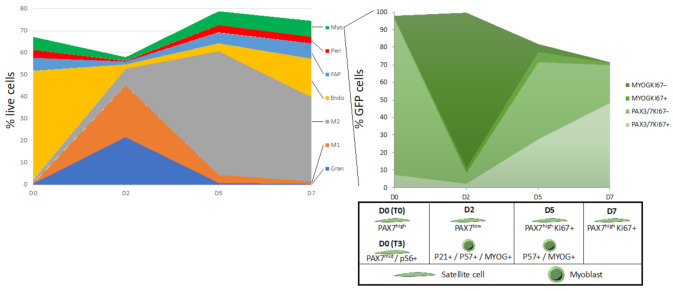
Cellular dynamics during muscle regeneration. (**Left**) Proportions of different cell populations before and following injury. (**Right**) Expression of key transcription factors in the myogenic lineage.

**Table 1 cells-10-00744-t001:** List of Antibodies Used in CyTOF Experiments.

Antigen	Localization	Metal Isotope	Company	Experiment
CD45	Surface	89Y	Fluidigm	Native and regenerating
LY6G	Surface	141Pr	Fluidigm	Native and regenerating
CASP3	Intracellular	142Pr	Fluidigm	Native and regenerating
P57	Intracellular	143Nd	Santa Cruz	Native and regenerating
PDGFRB	Surface	144Nd	Cell Signaling	Native and regenerating
F4/80	Surface	146Nd	Fluidigm	Native and regenerating
SMA	Intracellular	147Sm	Sigma	Native and regenerating
CD34	Surface	148Nd	BD Biosciences	Regenerating
PDGFRA	Surface	148Nd	Fluidigm	Native
p4EBP1	Intracellular	149Sm	Fluidigm	Native and regenerating
LY-6C	Surface	150Nd	Fluidigm	Native and regenerating
NESTIN	Intracellular	151Eu	Fluidigm	Native and regenerating
pAKT	Intracellular	152Sm	Fluidigm	Regenerating
CD3E	Surface	152Sm	Fluidigm	Native
CYCLINB1	Intracellular	153Eu	Fluidigm	Native and regenerating
SYNDECAN4	Surface	154Sm	Santa Cruz	Regenerating
ITGA7	Surface	154Sm	R&D Systems	Native
TCF4	Surface	155Gd	Fisher Sci	Native and regenerating
pSMAD1/5	Intracellular	156Gd	Cell Signaling	Native and regenerating
pSTAT3	Intracellular	158Gd	Fluidigm	Native and regenerating
P21	Intracellular	159Tb	Fluidigm	Regenerating
JAG1	Surface	159Tb	Santa Cruz	Native
PDGFRA	Surface	160Gd	Fluidigm	Regenerating
PAX3/7	Intracellular	161Dy	Santa Cruz	Native and regenerating
KI67	Intracellular	162Dy	Fluidigm	Native and regenerating
NG2	Surface	163Dy	Cell Signaling	Native and regenerating
LY-6A_E	Surface	164Dy	Fluidigm	Native and regenerating
CD31	Surface	165Ho	Fluidigm	Native and regenerating
pRB	Intracellular	166Er	Fluidigm	Native and regenerating
P27	Intracellular	167Er	Santa Cruz	Native and regenerating
MCADHERIN	Surface	168Er	Fisher Sci	Regenerating
CD8A	Surface	168Er	Fluidigm	Native
GFP	Intracellular	169Tm	Fluidigm	Native and regenerating
MYOGENIN	Intracellular	170Er	Santa Cruz	Native and regenerating
pERK1/2	Intracellular	171Yb	Fluidigm	Native and regenerating
CD11B	Surface	172Yb	Fluidigm	Regenerating
pS6	Intracellular	172Yb	Fluidigm	Native
VCAM	Surface	173Yb	BD Biosciences	Native and regenerating
MYF5	Intracellular	174Yb	Santa Cruz	Regenerating
NRF2	Intracellular	174Yb	Cell Signaling	Native
pHH3	Intracellular	175Lu	Fluidigm	Native and regenerating

## Data Availability

The CyTOF data files have been uploaded to FlowRepository (https://flowrepository.org/) under the following repository ids: FR-FCM-Z3L9 (early activation) and FR-FCM-Z3L8 (muscle regeneration).

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
