# Peer review of "Progressive and Coordinated Mobilization of the Skeletal Muscle Niche throughout Tissue Repair Revealed by Single-Cell Proteomic Analysis"

_cells, 2021, doi:10.3390/cells10040744_

Round 1

Reviewer 1 Report

In figure 4, there are some hollow circles instead of solid dots in the panel B, E, and F. Any specific meaning of these? What is NI here. Does that mean non-injured? Suggest to use D0, or UI (uninjured), which was described in the methods.

Either a list of abbreviations or defining when they first appear is suggested. There are a lot of abuse of the abbreviations. Remember the full name of each abbreviation needs to be provided when they first showed in the manuscript. For example, the word "tibialis anterior" appears five times and four of them followed by an abbreviation "(TA)". Only the first one is necessary and for the rest of them you can just use TA.

There are some other regulatory genes during muscle regeneration and repair such as MRF4 and Myf5. They work at the later stage of myogenesis. Did you see the change of the expression of these genes?

Author Response

In figure 4, there are some hollow circles instead of solid dots in the panel B, E, and F. Any specific meaning of these? What is NI here. Does that mean non-injured? Suggest to use D0, or UI (uninjured), which was described in the methods.

Indeed NI was an abbreviation for non-injured. We have updated the figure to keep the notation consistent throughout the paper. The uninjured data points are now also filled.

Either a list of abbreviations or defining when they first appear is suggested. There are a lot of abuse of the abbreviations. Remember the full name of each abbreviation needs to be provided when they first showed in the manuscript. For example, the word "tibialis anterior" appears five times and four of them followed by an abbreviation "(TA)". Only the first one is necessary and for the rest of them you can just use TA.

We have replaced all but two “tibialis anterior”s with “TA”. We chose to define it once in the main text, and once in the Methods section.

There are some other regulatory genes during muscle regeneration and repair such as MRF4 and Myf5. They work at the later stage of myogenesis. Did you see the change of the expression of these genes?

We did use a Myf5 antibody in our panel, but we saw no change in its expression. We also note that while preliminary FACS tests with the Myf5 antibody were promising, when it was coupled to heavy metals for CyTOF analysis, we never had strong signal. Porpiglia et al. 2017 used a Myf5 antibody in their CyTOF panel, and found it partially overlapping with MyoG in satellite cell subpopulations in resting muscle.

Reviewer 2 Report

In this study, Borok and colleagues developed a combination of mass cytometry and flow cytometry (CyTOF) panels to follow dynamic changes in the proteome of the main skeletal muscle population during muscle regeneration and early activation.

The study is based on ex vivo approach and all the experimental procedures are clearly stated. Hovewer, it lacks of a clear and original biological question.

Indeed, the relevance of this study is mainly technical, as their panel allows a deeper screening of the muscle regeneration process when performed in parallel with other tools. The final outcome is a not innovative work, whose results are not particularly striking and the main conclusions not fully supported by the data. The authors themselves in fact state that “the changes in different cell populations were in good agreement with previous reports using histology, FACS, scRNAseq and CyTOF”.

Overall this work might be much more appropriate for a method journal but providing some advance  with respect of what has been already done.

Major points:

  1. Figure 2C-D: the data are expressed as % of CD45+ or CD45- cells, not in % of events. This alters the interpretation of the data since it doesn’t consider the high variance in the relative percentages of these two cell groups between different time points. As such, it not possible to compare different time points in the same group of cells, but only the same time point among different cell types. For this reason, many of the conclusions on the cell proportions over time drawn from these graphs are not appropriate.
  2. The authors claim that their results (Figure 4B) support the idea that the regeneration of the entire muscle can be achieved from a small number of satellite cells. However, this conclusion cannot be drawn from their data. The decrease observed in the total PAX7+ cells at 2 days after injury is probably, mainly, due to their early activation, which is part of the regenerative process that these cells are sustaining. By day 7 the number of PAX7+ SCs has increased again due to the proliferation process, restoring the initial pool of PAX7+ SCs needed for the next rounds of regeneration. It would be more precise to state that a smaller fraction of SCs can replenish the pool.
  3. Figure 4G-H: the graphs are redundant. How do the authors explain the small amount of PAX7+/MYOD+ and MYOD+/MYOG+ cells?
  4. The authors conclude that “the most drastic changes in terms of cellular components of the niche occur in the first week following injury”. However, they analyzed samples only up to 7 days after damage. Their data show that many changes are still ongoing at the last time point analyzed. To draw this conclusion they should have included also later timepoints.
  5. General observation: the conclusions are not strongly supported by the authors’ data. For example, they did not definitively demonstrate, as they claim, that “only 20% of the muscle stem cell pool is sufficient to repair the injured muscle”.

Minor points:

  1. The Single tibialis anterior muscle dissociation and immunostaining for flow cytometry analysis is not formatted properly. I suppose this should be part of a dedicated method section
  2. Figure 5: the image is not complete and poorly intuitive. I would suggest to propose an other way to schematize the results obtained.

https://deliveroo.it/share/b1020e1b-9da8-4e0a-9533-0fad1732e987

Author Response

Major points:

  1. Figure 2C-D: the data are expressed as % of CD45+ or CD45- cells, not in % of events. This alters the interpretation of the data since it doesn’t consider the high variance in the relative percentages of these two cell groups between different time points. As such, it not possible to compare different time points in the same group of cells, but only the same time point among different cell types. For this reason, many of the conclusions on the cell proportions over time drawn from these graphs are not appropriate.

The reviewer raises a very valid point. We chose to represent the data in this way because the massive influx of immune cells (approximately 4% of live cells without injury, and more than 80% of live cells 5 days post-injury) drastically skews the proportions of other cell types. We have changed the wording of this section to stress that the proportions are relative. Additionally, Figure 5 represents most of the same population data relative to the total live cells.

  1. The authors claim that their results (Figure 4B) support the idea that the regeneration of the entire muscle can be achieved from a small number of satellite cells. However, this conclusion cannot be drawn from their data. The decrease observed in the total PAX7+ cells at 2 days after injury is probably, mainly, due to their early activation, which is part of the regenerative process that these cells are sustaining. By day 7 the number of PAX7+ SCs has increased again due to the proliferation process, restoring the initial pool of PAX7+ SCs needed for the next rounds of regeneration. It would be more precise to state that a smaller fraction of SCs can replenish the pool.

We agree with the reviewer and have added a sentence to address the contribution of early activation to the change in PAX7+ cell numbers: “However, this may also be due to early activation, rather than loss of the satellite cells.”

  1. Figure 4G-H: the graphs are redundant. How do the authors explain the small amount of PAX7+/MYOD+ and MYOD+/MYOG+ cells?

We agree that there is some potential overlap between these panels, and have thus moved the final panel to a supplemental figure. In our previous paper (Gattazzo et al. 2020), we showed that PAX7+/MYOD+ cells are a significant cell population during postnatal development, but this population is quite rare by postnatal day 49 and, based on our data, does not reach appreciable numbers at the regeneration time points we analyzed. We are not sure of the exact reason for this. We have now moved the MYOD+/MYOG+ data into a supplementary figure.

  1. The authors conclude that “the most drastic changes in terms of cellular components of the niche occur in the first week following injury”. However, they analyzed samples only up to 7 days after damage. Their data show that many changes are still ongoing at the last time point analyzed. To draw this conclusion they should have included also later timepoints.

This is a fair point, and we have changed the wording to “many drastic changes” instead of “the most drastic changes”.

  1. General observation: the conclusions are not strongly supported by the authors’ data. For example, they did not definitively demonstrate, as they claim, that “only 20% of the muscle stem cell pool is sufficient to repair the injured muscle”.

We have now done our best to taper back unsupported claims, including modification of the cited sentence: “This demonstrates that satellite cell loss and early activation significantly reduces the quiescent muscle stem cell pool, without impairing muscle regeneration.”

Minor points:

  1. The Single tibialis anterior muscle dissociation and immunostaining for flow cytometry analysis is not formatted properly. I suppose this should be part of a dedicated method section

The formatting has been corrected.

  1. Figure 5: the image is not complete and poorly intuitive. I would suggest to propose an other way to schematize the results obtained.

We agree that the Figure could be more aesthetically pleasing, but we find it to be an informative representation of the data. It also represents the cell proportions relative to the percentage of live cells. We have now also added the pericyte data to the figure.

Round 2

Reviewer 2 Report

The authors addressed the main technical issue raised by this Reviewer